# Efficient biomedical image segmentation on Edge TPUs

**Andreas M. Kist**[1,2]

ANDREAS.KIST@FAU.DE

[1] *Department of Artificial Intelligence in Biomedical Engineering, Friedrich-Alexander-University Erlangen-Nürnberg, Germany*

[2] *Division of Phoniatrics and Pediatric Audiology, Department of Otorhinolaryngology, Head- and Neck surgery, University Hospital Erlangen, Friedrich-Alexander-University Erlangen-Nürnberg, Germany*

**Michael Döllinger**[2]

MICHAEL.DOELLINGER@UK-ERLANGEN.DE

## Abstract

Biomedical semantic segmentation is typically performed on dedicated, costly hardware. In a recent study, we suggested an optimized, tiny-weight U-Net for an inexpensive hardware accelerator, the Google Edge TPU. Using an open biomedical dataset for high-speed laryngeal videoendoscopy, we exemplarily show that we can dramatically reduce the parameter space and computations while keeping a high segmentation quality. Using a custom upsampling routine, we fully deployed optimized architectures to the Edge TPU. Combining the optimized architecture and the Edge TPU, we gain a total speedup of $>79\times$ compared to our initial baseline while keeping a high accuracy. This combination allows to provide immediate results at the point of care, especially in constrained computational environments.

**Keywords:** Edge TPU, image segmentation

## 1. Introduction

Semantic segmentation is an important tool in biomedical data analysis. Many quantitative measures rely on the extracted information, such as cancer size, fetal development or voice quality. The latter relies on the semantic segmentation of the glottal area (Figure 1(a)) that has been an active area of research for years (Andrade-Miranda et al., 2020), only recently deep convolutional neural networks (DNNs) were applied (Gómez et al., 2020). However, the focus has been mainly on the segmentation quality and not particularly on the segmentation speed. In a recent study that we briefly present here, we showcase efficient DNNs that are both, fast and highly accurate (Kist and Döllinger, 2020), especially in constrained, CPU-only environments. We not only utilize computational tweaks similar to previous studies (Tan and Le, 2019), but also make use of a novel, inexpensive hardware accelerator termed Edge TPU (Cass, 2019). The use of Edge TPUs in biomedical tasks has been barely touched, therefore, we briefly present our advances on this topic.

## 2. Methods

We implement a U-Net and the DeepLabV3+ (Chen et al., 2017) architecture as described previously (Kist and Döllinger, 2020) in TensorFlow/Keras in v.1.15. Not supported operations by the Edge TPU were replaced accordingly, such as dilation in Conv2D layers. We

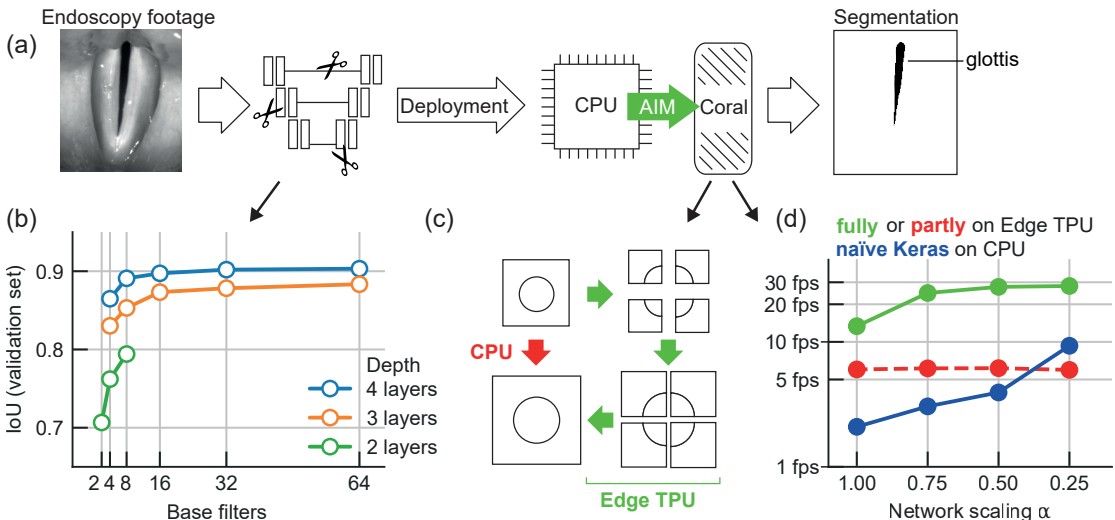

Figure 1: Biomedical semantic segmentation on Edge TPUs. (a) Glottis segmentation task with optimization strategies and deployment scheme. Aim is to map all operations to the Edge TPU. (b) Network performance across different U-Net depths and base filters. (c) Custom Upsampling2D operation. (d) Inference speed for naïve Keras on a conventional consumer CPU (blue), partly mapped with the default UpSampling2D operation to the Edge TPU (red) and using our custom UpSampling2D routine (green) across network size configurations.

further ensured that BatchNorm layers were not fused, as this caused issues in the compilation. We use the BAGLS dataset (Gómez et al., 2020) for semantic segmentation of the glottis (binary pixel-wise classification). We train our networks with the Adam optimizer, a Dice loss, a cyclic learning rate between $10^{-3}$ and $10^{-6}$, and a varying base filter size (Equation 1) to scale the U-Net size dynamically.

$$\text{filter}_{\text{Depth}} = \text{filter}_{\text{base}} \cdot 2^{\text{Depth}-1}, \text{Depth} \in [1, \text{Depth}_{max}] \tag{1}$$

Images were rescaled to 512×256 px. All networks were trained in quantization aware mode and subsequently converted to TFLITE. Next, the Edge TPU compiler mapped the operations to CPU and Edge TPU. When activation maps in the decoder exceeded a given size, we applied a custom UpSampling2D routine that allowed us to fully map all operations to the Edge TPU in contrast to the default UpSamling2D operation (see Figure 1(c)). We used the Intersection over Union (IoU) score as evaluation metric.

## 3. Results

We found that a variety of U-Net configurations are able to tackle the glottis segmentation task (Figure 1(a)) as measured by the IoU score on the validation dataset (Figure 1(b)). Our results suggest that the base filter count is more tolerated than the depth of the

network, indicating the importance of high-level features. Reducing the base filter count and using separable convolutions had the most impact on reducing the parameter space ($>99\%$ parameter reduction). We further found that Conv2Ds are more efficiently implemented in Edge TPUs than separable Conv2D layers similar to previous reports.

We encountered that upsampling large activation maps are partly mapped to the CPU, introducing a bottleneck across different network size configurations, here shown for the DeeplabV3+ architecture with the MobileNetV2 backbone (Figure 1(d)). However, using a custom UpSampling2D routine (Figure 1(c)), we were able to circumvent this issue. This affected the inference speed significantly and removed the CPU bottleneck (Figure 1(d)). In detail, we observed an inference speed of up to 30 fps compared to the constant 6 fps on the partially mapped configuration. In comparison, the same networks in the naive Keras environment have a runtime performance between 2 and 10 fps (Figure 1(d)), resulting in a 3-8× speed-up. Additionally, we do not observe significant drops in segmentation quality by int8 conversion (Kist and Döllinger, 2020). Combining the architecture optimization and the use of the Edge TPU, we gained in total a $>79\times$ speed-up compared to our initial baseline (Kist and Döllinger, 2020).

## 4. Conclusion

In this work, we highlight the use of Edge TPUs across different architectures in biomedical image segmentation and show that slight modifications to the architectures result in significant performance boosts. We envision that these findings will influence the deployment of DNNs in constrained or remote environments, such as endoscopic imaging setups.

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
