# OpenReview forum: "Efficient biomedical image segmentation on Edge TPUs"
_MIDL.io/2021/Conference/Short — MIDL 2021 Poster_

### Official Review · Reviewer_Y37x · 2021-04-30

**Confidence:** 4
**Final Rating:** 3

**Summary:**

This contribution is a short summary of an IEEE Access paper that targets efficient inference for endoscopic videos. The authors trained U-nets "or" DeepLabV3+ on the public BAGLS dataset on semantic segmentation of the glottis. Apart from reducing the activation maps to 8 bit values, the authors made several adaptations to enable inference on Google Edge TPUs. A few results indicate that inference at 30 FPS becomes possible with a careful implementation.

**Strengths:**

The authors have previously published the dataset, which greatly facilitates reproducing this and related work. The techniques and results look promising and interesting for the MIDL audience. The glottis segmentation task is relevant for voice disorders and voice research and justifies real-time segmentation techniques.

**Weaknesses:**

My main concern about the paper is that both methods and quantitative results are only briefly sketched. I am unsure whether to recommend to accept this MIDL contribution, because I believe the underlying work (as described in the IEEE Access paper) is highly relevant and interesting to the MIDL audience, or whether to (weakly) recommend rejection because the adaptations necessary for the Edge TPU are not really described, and also the quantitative evaluation has been stripped down too much to be able to understand.

**Deanonymize Review:**

yes

**Detailed Comments:**

What does "We implement the U-Net *or* DeepLabV3+…" mean? Did you implement both ("and")?

Subject missing? "Not supported by the Edge TPU were adjusted."

Maybe you can reduce the references further, in order to make room for one or two more informative sentences.

**Justification Of The Rating:**

I believe the format ("summary of previously published work") makes it necessary to shorten so much that the short paper itself becomes hard to follow, and I think a few improvements could alleviate the issues I saw, but the underlying methods and evaluations are definitely worth presenting and discussing at MIDL.

**Paper Type:**

both

**Special Issue:**

no

---

### Meta-Review · Area_Chair_TPsz · 2021-05-09

**Recommendation:** Accept (Poster)
**Confidence:** 4

**Metareview:**

The paper summarises an interesting journal paper about Edge-TPU computing. The reviewer points out that the underlying publication is of high interest but criticises that the reduction to three pages reduced the readability. I agree with them and recommend acceptance, but would also encourage the authors to modify where possible the paper that it is better to understand.

---

### Decision · Program_Chairs · 2021-05-11

Accept (Poster)